# Bentall Operation: Early Surgical Results, Seven-Year Outcomes, and Risk Factors Analysis

**DOI:** 10.3390/ijerph20010212

**Published:** 2022-12-23

**Authors:** Paolo Nardi, Calogera Pisano, Carlo Bassano, Fabio Bertoldo, Alessandro Cristian Salvati, Dario Buioni, Daniele Trombetti, Laura Asta, Mattia Scognamiglio, Claudia Altieri, Giovanni Ruvolo

**Affiliations:** 1Cardiac Surgery Division, Tor Vergata University Hospital, Tor Vergata University, 00133 Rome, Italy; 2Cardiology Unit of the Cardiac Surgery Division, Tor Vergata University Hospital, Tor Vergata University, 00133 Rome, Italy

**Keywords:** Bentall operation, ascending aorta replacement, aortic root surgery

## Abstract

**Aim:** To analyze early and mid-term outcomes of the Bentall operation. **Methods:** Two hundred and seventeen patients (mean age 65.6 ± 15.9 years, males/females 172/45) underwent Bentall operation in a 7-year period (January 2015–December 2021), on average, 30 Bentall operations occurred per year, using biological (n = 104) or mechanical (n = 113) valved conduits for the treatment of ascending aorta–aortic root aneurysms. Associate procedures were performed in 58 patients (26.7%); coronary artery bypass grafting (CABG) in 35 (16%). Mean follow-up was 55.2 ± 24 (median 60.2) months. Cox model analysis was used to assess risk factors, Kaplan–Meier and log-rank tests were used to assess different survival rates. **Results:** Operative mortality was 1.38%. At 7 years, survival, freedom from cardiac death, and event-free survival were 93% ± 2%, 99% ± 1%, and 81% ± 5%. NYHA class (*p* < 0.0001), trans-aortic valve mean (*p* < 0.0001) and maximum (*p* < 0.000) gradients, left ventricular hypertrophy (*p* < 0.05), and pulmonary arterial pressure (*p* = 0.002) significantly improved vs. preoperative values. Concomitant CABG during Bentall operation independently affected late outcomes (HR 1.9–2.3; *p*-values < 0.05). Late survival was affected by concomitant CABG (84% ± 8% vs. 95% ± 2%, *p* = 0.04), preoperative myocardial infarction (91% ± 9% vs. 97% ± 2%, *p* = 0.02), and biological vs. mechanical prostheses valved conduits (91% ± 9% vs. 95% ± 3%, *p* = 0.02). Event-free survival also was affected by concomitant CABG (62% ± 14% vs. 85% ± 5%, *p* = 0.005) and biological prostheses (78% ± 8% vs. 84% ± 6%, *p* = 0.06). Freedom from endocarditis–redo operation was 83% ± 9% for biological prostheses vs. 89% ± 6% for mechanical prostheses (*p* = 0.49). **Conclusions:** Low rates of operative mortality and late complications make Bentall operation the gold standard for the treatment of ascending aorta–aortic root aneurysms. Coronary ischemic disease affects late outcomes. Biological prostheses should be preferred for the elderly.

## 1. Introduction

Surgical treatment of ascending aortic aneurysms is still considered a complex at-risk procedure. When the aortic root is involved, Bentall–De Bono operation, initially described in 1968 [1], is the most performed operation. During the years, this procedure underwent important modifications, including the abandonment of the wrap-inclusion technique in favor of the coronary-button technique, and it has become a widely adopted standard of treatment of various root pathologies. In fact, although valve-sparing procedures are often the preferred approach in selected cases, Bentall operation remains universally applicable in an all-comer cohort and is not limited to favorable valve morphology, including patients at high risk of aortic complications, i.e., genetic family syndromes, bicuspid valve, altered root geometry, coronary ostia dislocation. The complete aortic root replacement can be performed using biological or mechanical valved conduits [2,3,4].

Even in experienced hands, the perioperative mortality remains not insignificant. In recent years, groups focused on aortic disease have reported 4–5% mortality rate, a not-negligible incidence of bleeding, and major cardiac and noncardiac postoperative complications, so much so that it has been proposed to define a risk score, already present for coronary and valve surgery, for surgical procedures for the treatment of ascending aorta aneurysms as well [2,3,5].

Furthermore, different study groups have focused on Bentall follow-up results with the use of mechanical and biological prostheses, taking into account that for mechanical prosthesis the main problem is the lifelong need for anticoagulant therapy, while the main benefit is the durability, and for biological prosthesis, the main problem is represented by the structural deterioration over time.

The aim of our study was to evaluate early and mid-term results of the modified Bentall operation in accordance with the routinely used coronary “button technique”, also focusing the analysis on surgical technical aspects that could improve early outcomes, especially by reducing postoperative complications, i.e., early postoperative bleeding, or the incidence of development of pseudoaneurysms due to coronary ostia stretch. We reviewed the results of both isolated and more complex Bentall operations, in particular, those associated with coronary artery bypass grafting. We also investigated the risk factors and the impact of the use of biological or mechanical prosthesis valved conduits on seven-year follow-up outcomes.

## 2. Materials and Methods

### 2.1. Patients and Methods

From January 2015 to December 2021, at the Cardiac Surgery Division of the Tor Vergata University Hospital, two hundred and seventeen patients (172 males, 45 females; mean age 65.6 ± 15.9 years) out of 238 (91%) elective procedures performed to treat ascending aorta expansive aneurysms underwent Bentall operation, with an average of 30 Bentall operations per year. The average percentage value of the EuroScore-2 was 5.0% ± 3.8%. Main surgical indication was performed for expansive ascending aorta (mean diameter 50.0 ± 7.3 mm) and aortic root (mean diameter 44.5 ± 6.6 mm) aneurysms. Moderate to severe degree of aortic valve insufficiency or steno-insufficiency was present in 124 patients (57%). One hundred and sixteen patients (53.5%) had tricuspid aortic valve, 87 (40%) bicuspid valve, and 14 (6.5%) were affected by genetic syndromes (Marfan syndrome, 13 patients; Loeys–Dietz syndrome, 1 case). Associated procedures, including coronary artery bypass grafting (CABG), mitral and tricuspid valve surgery, closure of the foramen ovale patency or atrial septal defect, and aortic arch repair, were performed in 58 patients (26.7%); of them, concomitant CABG was the most frequently performed associated procedure (n = 35, 16%). In the same period of surgical activity, 67 patients underwent emergency Bentall operation for acute aortic syndrome (n = 64) and for acute endocarditis in heart failure phase (n = 3). This group of patients was excluded from our study.

Surgical indications for the treatment of ascending aortic aneurysms were based on the anatomic, clinical, and etiological criteria, and on intraoperative findings, i.e., significant coronary ostia dislocation, aortic wall thickness, evidence of cardiac muscle in transparency at the level of right or noncoronary sinuses, and asymmetric dilation of valsalva sinus/sinuses.

The study was conducted according to the guidelines of the Declaration of Helsinki and approved by the Independent Ethics Committee of the Tor Vergata University Polyclinic (237.22). All patients gave their informed surgical consent. The study was designed to be a retrospective one.

### 2.2. Data Analysis

Data analysis included patient history with major cardiovascular risk factors: hypertension, dyslipidemia, smoking, diabetes, obesity, ischemic heart disease, atrial fibrillation, and physical examination, with evaluation of pre- and postoperative echocardiographic parameters. Operative mortality included death in hospital after operation at any time, or within 30 days after discharge. We also investigated the incidence of postoperative low cardiac output syndrome combined with acute kidney injury, the incidence of postoperative noncardiac complications, need for early surgical revision for bleeding, and permanent pacemaker implantation. 

Follow-up was obtained by reviewing the medical record and telephone interview with patients on clinical conditions, NYHA functional class, management of INR, on complications arising during follow-up, and on echocardiographic reports. Mean duration of the follow-up was 55.2 ± 24 (median 60.2) months after surgery. Adverse events were classified according to the definitions established by the Society of Thoracic Surgeons and the American Association for Thoracic Surgery “*Guidelines for reporting morbidity and mortality and cardiac valve interventions*” [6]. Follow-up was closed on July 2022, and was 99% complete; two patients were lost. 

### 2.3. Surgical Treatment 

Surgery was performed through full sternotomy in all cases. Once cardiopulmonary bypass was started, after ascending aorta cross-clamping at the level of the proximal region of the arch immediately upstream of the origin of the anonymous trunk, cardiac arrest was achieved using antegrade intermittent warm blood cardioplegia or St. Thomas cold crystalloid solution [7]. Modified Bentall operation was performed using composite graft valved conduits with mechanical prostheses (n = 113) (Abbott St. Jude Medical Inc., St. Paul, MN, USA, CarboSeal, CarboMedics Inc., Austin, TX, USA; Corcym) or using a tubular cylinder graft (Intervascular, Datascope Corp., Wayne, NJ, USA) sutured to the biological prostheses (Perimount Carpentier, St. Jude Trifecta) (n = 104). Surgical technique consisted of proximal implantation of prosthetic valved conduit, reimplantation of the left coronary ostium, distal aortic anastomosis with Teflon reinforcement, and reimplantation of the right coronary ostium. The coronary ostia were reimplanted with the “button technique” using a 5-0 polypropylene suture. In the presence of a thinned coronary ostia wall, the suture of the left and right coronary ostium to the cylindrical prosthesis was reinforced in the perianastomotic site with a small strip of Teflon. Particular care was taken during the suturing of the coronary ostia on the tubular prosthesis and during the distal suture on the aortic wall, developing the correct tension of the polypropylene thread, using a hook and gently pulling the thread. 

### 2.4. Statistical Analysis

Statistical analysis was performed with the use of Stat View 4.5 (SAS Institute Inc., Abacus Concepts, Berkeley, CA, USA). All continuous values were expressed as mean plus or minus one standard deviation of the mean. The chi-squared or the Fisher’s exact test for categorical variables and the unpaired Student’s *t*-test for continuous variables were calculated to perform the univariate analysis to detect potential risk factors related with in-hospital mortality and cardiac complications, i.e., low cardiac output syndrome and acute kidney injury. Variables with a *p*-value less than 0.1 were included in the multivariate logistic regression analysis. Preoperative analyzed variables were sex, age, New York Heart Association class, body mass index, body surface area, rhythm, associated morbidity, i.e., arterial hypertension, smoking habit, dyslipidemia, diabetes mellitus, obstructive pulmonary disease, peripheral vascular disease, increased serum creatinine level, presence of aortic valve stenosis and/or insufficiency, dimension of expansive aortic aneurysm, and presence of ischemic heart disease, i.e., previous myocardial infarction. Echocardiographic variables measured were left ventricular ejection fraction, end-systolic and end-diastolic diameters, left atrium size, septum and posterior wall thickness, and systolic pulmonary artery pressure. Perioperative variables included the mean duration of cardiopulmonary bypass and aortic cross-clamp time, type of heart prostheses valve implanted, i.e., mechanical versus biological, and combined procedures, in particular CABG.

Late survival, event-free survival, and freedom from late cardiac death and from prosthetic valve-related events were calculated by the use of the Kaplan–Meier method; all measurements were expressed as mean values of percentage plus or minus 1 SD. The Mantel–Cox test was used to compare the curves of freedom from events, i.e., in patients who had undergone isolated Bentall operation versus Bentall plus CABG operation, and in those who had undergone mechanical versus biological prostheses implantation. The Cox proportional hazards method was used to estimate the influence of the analyzed variables on time to death. Echocardiographic variables obtained in the postoperative period and during the follow-up were compared with the preoperative ones. *p*-value of less than 0.05 was considered significant in all types of performed statistical analyses.

## 3. Results

Patients affected by ascending aorta aneurysms with a tricuspid aortic valve were older (70 ± 14 years) in comparison to those with a bicuspid aortic valve (64 ± 14 years) (*p* = 0.004) or affected by genetic syndromes (40.5 ± 14 years) (*p* < 0.0001). The mean age of implantation of valved conduits with mechanical prostheses was 56.7 ± 14.4 years, and the mean age of implantation of valved conduits with biological prostheses was 76.1 ± 10.0 years (*p* < 0.0001). As expected, cardiopulmonary bypass times (182 ± 58 min vs. 114 ± 26 min, *p* < 0.0001) and aortic cross-clamp (151 ± 26 min vs. 98 ± 22 min, *p* < 0.0001) times were longer in combined procedures. As expected, EuroScore-2 was higher in patients who had undergone Bentall plus CABG in comparison with isolated operations (6.6% ± 1.5% vs. 4.7% ± 0.3%, *p* = 0.05). In addition, patients who had undergone Bentall plus CABG had an older age at the time of the surgery (72.1 ± 13 years vs. 64.4 ± 16 years, *p* = 0.001), a higher incidence of arterial hypertension (90% vs. 75%, *p* = 0.04), and calcific steno-insufficiency on the tricuspid aortic valve (79% vs. 47%, *p* = 0.05). The mean number of coronary bypass graft per patient was 2 ± 1. Left internal mammary artery was used as a graft for the anterior descending artery when the latter was involved in atherosclerotic disease, and in all cases, the aim was to obtain complete revascularization. Operative mortality was 1.38% (n = 3): 0.62% (n = 1/159) for isolated Bentall operation, and 3.44% (n = 2/58) for combined procedures. There was no mortality in patients who had undergone mechanical prostheses valved conduit implantation. A 85-year-old male patient died from postoperative perioperative respiratory failure and septic shock; another 80-year-old male patient died from low cardiac output syndrome and multiple organ failure; the other one 78-year female patient died for sudden death at 3 weeks from operation. Low cardiac output syndrome occurred in 20 patients (9.22%). In the logistic regression analysis, preoperative higher value of pulmonary arterial pressure (43 ± 14 vs. 33 ± 7 mmHg), longer times of cardiopulmonary bypass (167 ± 63 min vs. 126 ± 43 min), and aortic cross-clamp (132 ± 46 min vs. 108 ± 33 min) were detected as independent predictors of operative mortality and cardiac complications (Table 1).

The incidence of postoperative early surgical re-exploration for bleeding, stroke, and pacemaker implantation were 3.22% (n = 7/217), 0.46% (n = 1/217), and 4.15% (n = 9/217), respectively. In four patients (1.84%), it was necessary to perform a temporary percutaneous tracheostomy due to severe primary respiratory insufficiency. The median postoperative in-hospital stay was 9.5 days.

### 3.1. Survival and Freedom Curves from Adverse Events

During follow-up, there were 10 deaths out of 212 patients (4.7%); of them, two patients died for cardiac causes, i.e., endocarditis and congestive heart failure. At seven years, actuarial survival was 93% ± 2% (Figure 1) and adverse-event-free survival was 81% ± 5% (Figure 2). Freedom from death from cardiac causes was 99% ± 1%, from endocarditis–redo operation, 86% ± 5%, and from stroke, 99% ± 1%, respectively (Figure 3a–c).

In the Cox regression analysis, concomitant CABG during the Bentall operation was the only independent predictor for reduced survival (HR = 1.9, *p* = 0.04) and event-free survival (HR = 2.3; *p* = 0.02). All the other variables mentioned in the statistical section of the methods, including age and the type of valve disease, i.e., steno-insufficiency or isolated insufficiency of the aortic valve, were not identified in the Cox analysis as risk factors for reduced survival and event-free survival. In the Mantel–Cox log-rank test, late survival was affected by CABG (84% ± 8% vs. 95% ± 2%, *p* = 0.04), preoperative myocardial infarction (91% ± 9% vs. 97% ± 2%, *p* = 0.02), and biological vs. mechanical prostheses valved conduits (91% ± 9% vs. 95% ± 3%, *p* = 0.02) (Figure 4a–c). Freedom from endocarditis–reoperation was 83% ± 9% for biological vs. 89% ± 6% for mechanical prostheses valved conduits (*p* = not significant) (Figure 5). Overall event-free survival also was affected by CABG (62% ± 14% vs. 85% ± 5%, *p* = 0.005) (Figure 6) and, although with a statistical value that does not reach significance, by the use of biological prostheses (78% ± 8% vs. 84% ± 6%, *p* = 0.06) (Figure 7).

### 3.2. Clinical Conditions during the Follow-Up Period

New York Heart Association functional class significantly improved in comparison with preoperative value (1.3 ± 0.5 vs. 2.0 ± 0.9; *p* < 0.0001). As compared with preoperative values, echocardiography data evaluation showed a significant improvement of cardiac parameters in terms of trans-aortic valve mean (*p* < 0.0001) and maximum (*p* < 0.0001) gradients, regression of the left ventricular hypertrophy (*p* < 0.05), and reduction of pulmonary arterial pressure (*p* = 0.002) (Table 2).

## 4. Discussion

The principal findings of our study are that today, again, Bentall operation can be considered the gold standard for the treatment of different types of ascending aortic root aneurysms being associated with low rates of operative mortality and late complications; concomitant coronary ischemic disease and myocardial infarction affect late outcomes; and biological prostheses should be preferred for the elderly. In our experience, in-hospital mortality was 0.62% for the isolated Bentall operation and was absent in younger patients, thus being comparable to the operative risk predicted by the STS risk calculator and by the EuroScore-2 in patients undergoing coronary artery bypass grafting or isolated aortic valve replacement. Our findings appear to be in agreement with what is reported in large case studies in the literature [8,9,10,11,12,13]. Even in the combined operations, the operative mortality, although approximately three times higher, i.e., 3.44%, was nevertheless acceptable, also taking into consideration the preoperative EuroScore-2 value, which was greater than 6% in this subgroup of patients. Furthermore, it is reported that the greater the number of procedures per year, the lower the operative mortality. In our center, Bentall operation is the most practiced for the treatment of ascending aortic aneurysms, accounting for over 90% of aortic interventions, with an average of 30 Bentall operations per year in the observed period of surgical activity. For this reason, over the years we have extended the surgical indication to include expansive aneurysms of 50 mm in the absence of risk factors in order to prevent acute aortic complications [14]. On the contrary, in low-volume centers, mortality can exceed 5%, and the incidence of major cardiac complications, such as myocardial infarction with or without low cardiac output syndrome, and complete AV block, can reach a range between 11 and 15% [2,3]. Similarly, the incidence of stroke, renal failure, and postoperative bleeding is not negligible, especially during combined interventions, reaching values, respectively, greater than 3%, 6%, and 9% [2,3,15]. In our analysis on the perioperative outcome, independent risk factors for early mortality and major cardiac complications, i.e., low cardiac output syndrome, were a higher value of preoperative pulmonary arterial pressure, i.e., >40 mmHg, and longer times of cardiopulmonary bypass and aortic cross-clamp. In several reports, independent predictors of mortality reported are age greater than 70 years, preoperative NYHA III-IV, reoperation, concomitant mitral valve replacement, and CABG, or a greater preoperative comorbidity. The mean duration of cardiopulmonary bypass, i.e., 114 ± 26 min, and aortic clamping, i.e., 98 ± 22 min, observed in our series was found to be in line with what was reported in a large meta-analysis of 29 studies, performed in 3298 patients, i.e., 185 (range 112 to 318) minutes and 124 (range 76 to 242) minutes, respectively [15]. The low operative mortality, especially in the isolated Bentall operation and in patients who implanted mechanical conduits, i.e., 0.62% and absent, respectively, may have depended on various intraoperative technical factors that we adopted. Most of the operations (187 out of 217, 86%) were performed by the director surgeon (G.R.), very experienced in the treatment of aortic pathology, who implemented special technical measures. The most fearful complications after Bentall operation are related to the malposition of the perianastomotic suture of the coronary ostia, which can cause torsion, dissection, laceration, or late development of pseudoaneurysm. An incorrect ostia reimplantation can cause mortality due to myocardial infarction, low cardiac output syndrome, and right ventricular dysfunction. In fact, particular care was taken by us, which consists of the partial mobilization of both ostia, in order to avoid postoperative kinking. Moreover, especially in order to prevent the development of late periostial pseudoaneurysms, the correct orientation of the suture of the ostia, in particular of the right one, was assessed by using colored reference point for the correct initiation of the anastomosis on the slot of the tubular Dacron prosthesis.

To prevent periostial coronary buttons bleeding, particular care was performed during the anastomosis, gently favoring the juxtaposition between the coronary ostia and prosthetic tissue, and at the same time pulling the polypropylene thread of the anastomosis for correct hemostasis with a hook after the first three–four steps with 5-0 polypropylene suture. Early bleeding from the implantation of the right coronary ostium was found in one patient only (0.46%), who underwent surgical re-exploration on the fourth postoperative day. We used the same technique to perform the distal anastomosis between the prosthetic conduit and the aortic tissue with the 4-0 polypropylene, with the interposition of a Teflon strip along the entire perianastomotic suture. Such technical measures could justify the fact that during the follow-up, we did not observe the development of perianastomotic coronary button pseudoaneurysms or at the distal level of the anastomosis of the prosthesis. On the contrary, an exception was made for the development of proximal abscess pseudoaneurysms due to prosthetic endocarditis observed in 10 patients during follow-up. Yamabe et al. [16], on a series of 580 patients undergoing modified Bentall operation, reported that at 10 years there was a 6.1% reoperation rate for noninfectious causes. Of the 34 reoperated patients, cause of reoperation was, in 20.6% of cases, the formation of pseudoaneurysms or aneurysms of the aorta distal to the prosthetic conduit, and, in 2.9% of cases, the development of coronary button pseudoaneurysm.

Considering the average age at surgery, i.e., 66 years, the 7-year actuarial survival of 93% and event-free survival of 89% (Figure 1 and Figure 2) were very satisfactory, and in line with what is reported in the literature. Di Marco et al. [11] reported a survival rate of 84.1% at 5 years and 65.5% at 10 years, respectively, on a population of 1045 patients with an average age at surgery of 59 years. Gaudino et al. [12], on 289 patients with a valved conduit with mechanical prostheses and 421 with biological prostheses, reported a survival of 91.7% and 81.5%, respectively, at 5 years; van Putte et al. [17], out of 528 patients with a mean age of 54 years, reported a 5-year survival rate of 87%. In the Cox regression analysis, the main independent predictor of reduced survival and freedom from adverse events was the concomitant presence of ischemic heart disease (HRs, 1.9–2.3). In fact, both the previous myocardial infarction and the need to perform concomitant CABG resulted in an overall reduction in freedom from adverse events of about 10–20% at 7 years (Mantel–Cox tests) (Figure 3b,c). Concomitant ischemic heart disease has also been recognized as a risk factor in other studies [11,12,13,17]. However, it should be emphasized that the freedom from cardiac death was very satisfactory, being 99% (Figure 2). It could be hypothesized that it was not the presence of ischemic disease only, “per se”, that determined higher risk of death related to cardiac causes, but rather the fact that patients were also affected by higher EuroScore risk profile and greater comorbidity, such as advanced age and risk cardiovascular factors that can lead to a higher all-causes mortality.

A final aspect that we analyzed was the choice of the type of valved conduit, with mechanical or biological prostheses. Our policy adopted for Bentall operation follows that for isolated aortic valve replacement, with the exception of the cases provided for by the guidelines or the patient-related choice for the implantation of a biological prosthesis, we prefer to use the biological prostheses for patients of 70 years, and the mechanical prostheses in younger patients with no contraindications to long-term anticoagulation. This orientation is related to the fact that even in the most recent 2021 guidelines, a higher mortality was observed during long-term follow-up, i.e., at 10 and 15 years, in patients under the age of 60 years, and aged between 50 and 70 years, who have an implanted biological valve in the aortic position [18]. Furthermore, during follow-up we did not observe major bleeding events after mechanical prostheses implantation, and freedom from stroke was 99%. For these reasons, from our analysis, we do not believe that the extensive use of biological prostheses for younger patients is justified, as the operative risk of reoperation may exceed that associated with long-term anticoagulant therapy. Moreover, the reported risk of redo operation is higher than 7%, and if the etiology is endocarditic, it reaches over 20% [16,19,20]. One reason for the low incidence of thromboembolic and hemorrhagic valve mechanical-related events may be found in the particular care that physicians and younger patients take in taking anticoagulant therapy at the correct dosage, to maintain the INR range between two and three. Several studies in the literature compared the results of Bentall with biological and mechanical conduit, analyzing survival, the risk of reoperation, and endocarditis. Svensson and coworkers [13] reported that bioprosthesis Bentall patients had higher risk of late death at 15 years (57% vs. 14–26% for other aortic root replacement procedures, *p* < 0.0001). Werner et al. [21], in patients aged 50–70 years, found at 10 years a survival rate not different between mechanical (n = 151) and biological (n = 110) valved conduits, showing a higher risk in the mechanical group in the early postoperative phase, which declined during the follow-up (*p* = 0.069). Pantaleo and coworkers [22], in a propensity score analysis of 138 mechanical and 138 biological Bentall operations, reported, at 5 years, similar survival (83.7% vs. 87.3%, *p* = 0.9), a substantially similar freedom from proximal aortic reintervention (99% vs. 93%) with a *p* value at the log-rank test of 0.07, and from endocarditis (99% vs. 83%, against biological prostheses, *p* = 0.2). Castrovinci et al. [15], in a meta-analysis of 14 studies on 1882 biological conduits implanted in patients aged 67 years old, reported at 5 years of follow-up a survival rate of 76%, freedom from reoperation at 90%, and freedom from endocarditis at 94%. In our analysis, freedom from endocarditis and reoperation at 6 years was substantially the same for the two types, mechanical (n = 113) and biological (n = 101) prostheses (93.6% vs. 93.2%, *p* = 0.5) (Figure 5), while actuarial survival (95% vs. 91%, *p* = 0.01) and event-free survival (89% vs. 88%, *p* = 0.06) (Figure 7) were better after mechanical prostheses implantation. A possible, or at least one, explanation of the reduced freedom from adverse events in patients with biological prostheses in the mid-term follow-up could be related to their older age at the intervention and to a greater comorbidity, i.e., related to the ischemic heart disease, although the statistical analysis did not detect the older age as a risk factor. Therefore, in the light of the comparative data observed during our follow-up and from other studies, and of the considerations we mentioned above, we would like to recommend the use of biological prostheses in older patients.

Despite being a fairly large study population, the more important limitations include its retrospective nature and a follow-up period up to the medium term.

In conclusion, low rates of operative mortality and late complications make Bentall operation the gold standard for the treatment of aortic root aneurysms. Coronary ischemic disease and myocardial infarction affect late outcomes. Biological prostheses should be preferred for the elderly.

## Figures and Tables

**Figure 1 ijerph-20-00212-f001:**
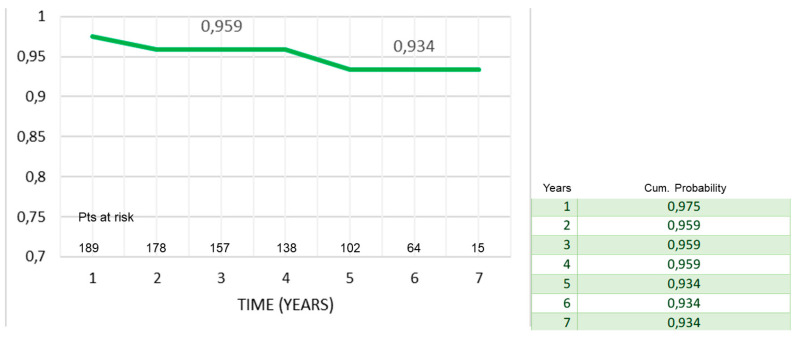
Survival after Bentall operation (mean follow-up, 55 ± 24 (M 60.2) months).

**Figure 2 ijerph-20-00212-f002:**
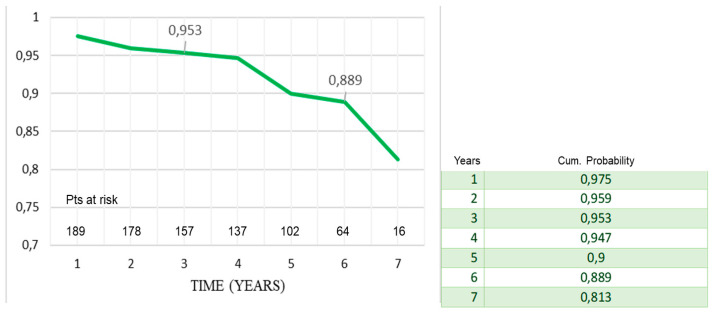
Event-free survival after Bentall operation.

**Figure 3 ijerph-20-00212-f003:**
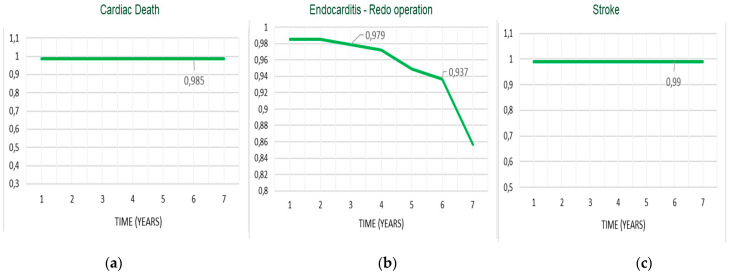
Freedom from late cardiac death (**a**); from endocarditis and redo operation (**b**); from stroke (**c**).

**Figure 4 ijerph-20-00212-f004:**
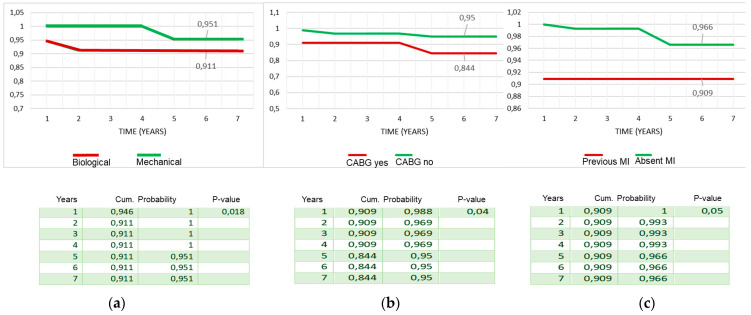
Survival stratified by biological vs. mechanical prostheses (**a**); concomitant CABG (**b**); presence of previous MI (**c**) (log rank, Mantel–Cox test). CABG: coronary artery bypass grafting; MI: myocardial infarction.

**Figure 5 ijerph-20-00212-f005:**
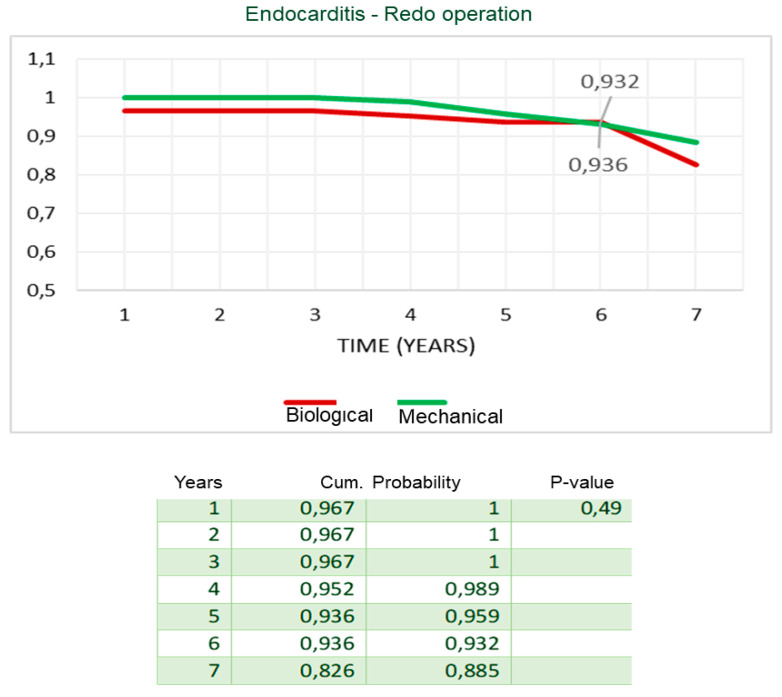
Freedom from endocarditis–redo operation stratified by biological vs. mechanical prostheses valved conduits (Mantel–Cox test).

**Figure 6 ijerph-20-00212-f006:**
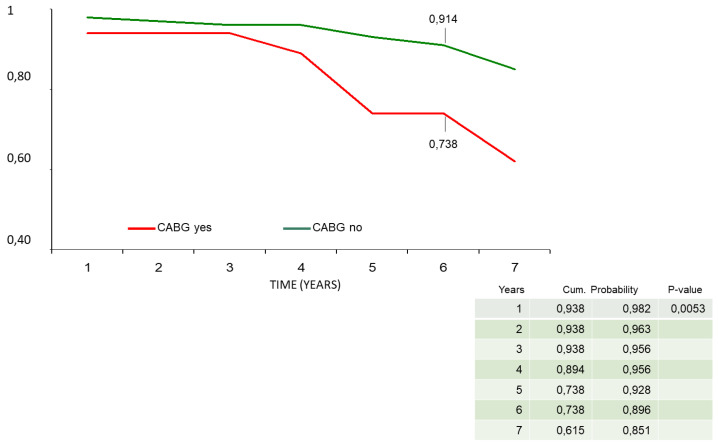
Adverse-event-free survival stratified by concomitant CABG (Mantel–Cox test). CABG: coronary artery bypass grafting.

**Figure 7 ijerph-20-00212-f007:**
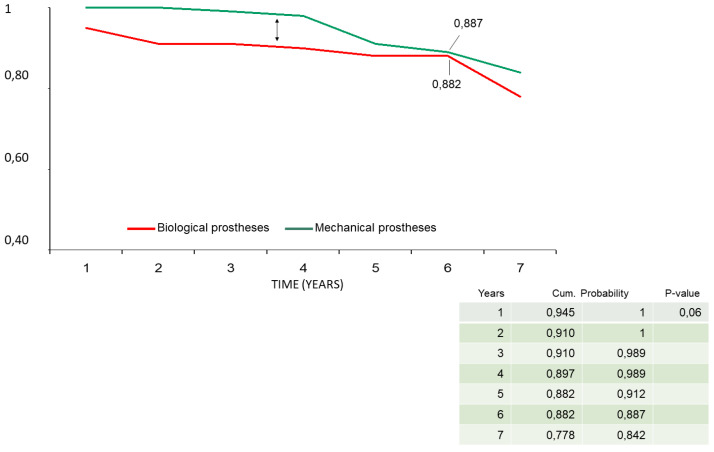
Event-free survival stratified by biological vs. mechanical prostheses valved conduits (Mantel–Cox test).

**Table 1 ijerph-20-00212-t001:** Independent predictors for hospital mortality and low cardiac output syndrome.

Variables	Hazard Ratio	95% CI	*p*-Value
Preoperative pulmonary arterial pressure	2.06	1.01–1.67	0.04
Longer CPB time	2.14	1.02–1.67	0.03
Longer aortic cross-clamp time	1.94	0.66–1.00	0.05
Age at operation			0.07
Concomitant CABG			0.89

CPB: cardiopulmonary bypass; CABG: coronary artery bypass grafting.

**Table 2 ijerph-20-00212-t002:** Echocardiographic variables.

Variables	Preoperative	Follow-up *	*p*-Value
Left ventricular end-diastolic diameter, mm	54.3 ± 9.7	53.0 ± 13	0.09
Left ventricular end-systolic diameter, mm	37.3 ± 8.1	35.7 ± 12	0.32
Left ventricular septum thickness, mm	13.0 ± 2.8	12.0 ± 1.1	0.04
Posterior wall thickness, mm	12.6 ± 2.7	10.9 ± 1.2	0.02
Left ventricular ejection fraction	0.55 ± 0.09	0.58 ± 0.07	0.17
Systolic pulmonary art. pressure, mmHg	34.3 ± 8.3	27.0 ± 4.8	0.002
Supra-coronary ascending aorta, mm	50.0 ± 7.3	27.0 ± 4.8	<0.0001
Aortic root, mm	44.5 ± 6.6	30.0 ± 1.7	<0.0001
Aortic valve peak gradient, mmHg	65 ± 31	27 ± 12	<0.0001
Aortic valve mean gradient, mmHg	44 ± 17	16 ± 7.4	<0.0001

* Not including deaths and patients lost during follow-up.

## Data Availability

Surgical and Clinical Data Base of the Tor Vergata University Hospital. Echocardiography data acquisition and interpretation: C.A.

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
