# Peer review of "Bentall Operation: Early Surgical Results, Seven-Year Outcomes, and Risk Factors Analysis"

_ijerph, 2022, doi:10.3390/ijerph20010212_

Round 1

Author Response

  • To Reviewer 1: we thank very much the Reviewer 1 for his positive evaluation of the manuscript. In the text, we have reported in the Results section that statistical analysis did not predict the type of aortic valve pathology as risk factor for increased mortality during follow-up (see in red colour in the text).

Moreover, we have reported in details the preoperative results of the characteristics of patients undergone Bentall operation plus CABG in the Results section (see in red colour).

At the echocardiographic exams during we did not observe structural valve deterioration of biological prostheses implanted, but these data, as you can see, are related to a mid-term time period and not to a long-term follow-up. The need for redo operation was done based on the secondary valve deterioration due to endocarditis prosthetic valve infection in all cases.

Thank you again.

Reviewer 2 Report

It is highly recommended to include more scientific component in such an expansive interesting clinical material.

1.       The aim of the paper is to substantiate the effectiveness of Bentall operation in various groups of patients. The main contribution is determined by the fact that the dependence of postoperative long-term survival on concomitant coronary artery disease, which required surgical correction of CABG, was revealed.

2.       It would be additionally interesting if groups of patients with Bentall operation for aortic stenosis and aortic insufficiency, as well as for the combination of these pathologies with coronary artery disease, were analyzed.

Strengths: of particular interest is the use of bioprosthetic valved conduits and the state of biological tissues in the long term: degradation, calcification, the need for redo operation.

3.       The figures are correct, the tables are clear to understand.

Author Response

  • Reviewer 2: we thank very much the Reviewer 2 for his suggestions and comments to implement the manuscript. In the text, at the materials and Methods section we have reported the number of Bentall operation performed in the same surgical period of activity in the emergency setting (see in red colour).

Age at operation was not identified as predictive risk factor for increased mortality during follow-up (obviously, it is a follow-up of a medium term period of observation, and not for a long-term period over 10 years…). We did not perform a propensity matched analysis.

More data in detail of patients undergone Bentall operation plus CABG have been reported in the Results section (see in red colour) as required by you. The use of LIMA graft for LAD was always used in patients affected by atherosclerotic significant ischemic disease of the left anterior descending artery, and complete revascularization was always achieved base on the extension of the coronary ischemic disease.

Round 2

Reviewer 1 Report

The authors have clarified some points.

In the method section inclusion is described as elective surgery. However it is stated, that only the last of three groups has been excuded. In the case all patients with acute aortic syndrom have been included, the term elective surgery is incorrect.

It is general agreement, that biologic prostheses should be preferred in elderly patients. However as endocarditis is higher, thromboembolic events are equal and survival is lower there is no proof of this by the results. The conclusions thus have to be adapted.

Author Response

  • To Reviewer 1: we thank very much the Reviewer 1 for further revision of the manuscript. Regarding the first question, we apologize for the error in the text but we mean 3 patients with acute endocarditis and NOT 63. Therefore, Bentall operatio in emergency setting were excluded from our study (see in red colour in the text, n=3).

Moreover, we have adapted the conclusion of the manuscript in the last part of Discussion section (see in red colour).
